# Comparison of Lung Ultrasound versus Chest X-ray for Detection of Pulmonary Infiltrates in COVID-19

**DOI:** 10.3390/diagnostics11020373

**Published:** 2021-02-22

**Authors:** María Mateos González, Gonzalo García de Casasola Sánchez, Francisco Javier Teigell Muñoz, Kevin Proud, Davide Lourdo, Julia-Verena Sander, Gabriel E. Ortiz Jaimes, Michael Mader, Jesús Canora Lebrato, Marcos I. Restrepo, Nilam J. Soni

**Affiliations:** 1Department of Internal Medicine, Hospital Universitario Infanta Cristina, 28981 Parla, Madrid, Spain; ma.mateosglez@gmail.com (M.M.G.); ggcasasolaster@gmail.com (G.G.d.C.S.); javier.teigell@gmail.com (F.J.T.M.); davide.tdsco@gmail.com (D.L.); 2Department of Medicine, Complutense University, 28040 Madrid, Spain; 3Section of Pulmonary and Critical Care Medicine, South Texas Veterans Health Care System, San Antonio, TX 78229, USA; proud@uthscsa.edu (K.P.); ortizg3@uthscsa.edu (G.E.O.J.); restrepom@uthscsa.edu (M.I.R.); 4Division of Pulmonary Diseases & Critical Care Medicine, University of Texas Health San Antonio, San Antonio, TX 78229, USA; 5Médicins Sans Frontières, 08005 Barcelona, Spain; Julia.Sander@barcelona.msf.org; 6Research and Development Service, South Texas Veterans Health Care System, San Antonio, TX 78229, USA; Michael.Mader2@va.gov; 7Department of Internal Medicine, Fuenlabrada University Hospital, 28942 Fuenlabrada, Madrid, Spain; jesuscanoralebrato@gmail.com; 8Division of General & Hospital Medicine, University of Texas Health San Antonio, San Antonio, TX 78229, USA

**Keywords:** ultrasound, imaging, X-ray, chest, diagnosis, SARS

## Abstract

Point-of-care lung ultrasound (LUS) is an attractive alternative to chest X-ray (CXR), but its diagnostic accuracy compared to CXR has not been well studied in coronavirus disease 2019 (COVID-19) patients. We conducted a prospective observational study to assess the correlation between LUS and CXR findings in COVID-19 patients. Ninety-six patients with a clinical diagnosis of COVID-19 underwent an LUS exam and CXR upon presentation. Physicians blinded to the CXR findings performed all LUS exams. Detection of pulmonary infiltrates by CXR versus LUS was compared between patients categorized as suspected or confirmed COVID-19 based on reverse transcriptase-polymerase chain reaction. Sensitivities and correlation by Kappa statistic were calculated between LUS and CXR. LUS detected pulmonary infiltrates more often than CXR in both suspected and confirmed COVID-19 subjects. The most common LUS abnormalities were discrete B-lines, confluent B-lines, and small subpleural consolidations. Most important, LUS detected unilateral or bilateral pulmonary infiltrates in 55% of subjects with a normal CXR. Substantial agreement was demonstrated between LUS and CXR for normal, unilateral or bilateral findings (Κ = 0.48 (95% CI 0.34 to 0.63)). In patients with suspected or confirmed COVID-19, LUS detected pulmonary infiltrates more often than CXR, including more than half of the patients with a normal CXR.

## 1. Introduction

Diagnosing coronavirus disease 2019 (COVID-19), the disease caused by the novel coronavirus SARS-CoV2, has been a major challenge as the pandemic has spread rapidly across the globe. Most patients present with nonspecific symptoms, including fever, cough, dyspnea, myalgias, and headache [1], that are indistinguishable from other respiratory infections. To confirm the disease in suspected patients, clinicians most often order reverse transcriptase-polymerase chain reaction (PCR) testing, but PCR testing has limited availability, relatively high false negative rates early in the course of the disease, and a delay of a few hours to days for results to be obtained [2,3].

Diagnostic imaging is being used to support a diagnosis of COVID-19 by detection of pulmonary infiltrates in suspected patients. Chest computed tomography (CT) scans have demonstrated superior diagnostic sensitivity for detecting pulmonary infiltrates in COVID-19 compared to chest X-ray (CXR) with reported sensitivity of 97–98% after 6 days of symptoms [2,3,4,5]. Though sensitive for pulmonary infiltrates, obtaining chest CT scans in all suspected COVID-19 patients is impracticable due to limited access to CT scanners worldwide and infection control requirements for disinfecting CT scanners. The American College of Radiology has recommended against routine use of CT scans for evaluating patients with suspected COVID-19 [6]. For these reasons, CXR and lung ultrasound (LUS) have been the primary imaging modalities used in the diagnosis of COVID-19 worldwide. CXRs can be obtained rapidly with minimal radiation exposure to patients, but have low sensitivity (46–69%) for detecting pulmonary infiltrates in COVID-19 patients [7,8].

Lung ultrasound (LUS) is an attractive alternative to CXRs and CT scans in COVID-19. Point-of-care or bedside LUS has several unique advantages in COVID-19, including immediate availability of findings to guide clinical decision-making, availability of portable ultrasound devices in austere settings such as field hospitals, repeatability to monitor patients serially, and ease of machine decontamination. Studies in non-COVID-19 patients have shown LUS has superior sensitivity (95% (95% CI 92–96%) vs. 49% (40–58%)) and similar specificity (94% (CI 90–97%) vs. 92% (CI 86–95%)) compared to CXRs when using chest CT scan as the gold standard [9]. Several recent studies have described lung ultrasound patterns in COVID-19 [10,11,12,13,14,15,16], but few studies have compared the diagnostic accuracy of LUS versus CXR for identifying lung abnormalities [17,18]. The objective of this study was to assess the correlation of LUS and CXR for detecting pulmonary infiltrates in COVID-19 patients.

## 2. Materials and Methods

### 2.1. The Study Design and Subjects

A prospective observational study of consecutive patients presenting with a clinical diagnosis of COVID-19 during the first COVID-19 surge in Spain was conducted from March 18, 2020 to April 5, 2020. The setting was an emergency department of a 247-bed university-affiliated teaching hospital in Madrid, Spain. Subjects were eligible for enrolment if they were an adult (age >18 years) and had a clinical diagnosis of COVID-19 based on classic symptoms of COVID-19 (fever, chills, cough, shortness of breath, sore throat, headache, myalgias, anosmia, ageusia, or diarrhea), close contact with an individual with active COVID-19, and abnormal laboratory findings (lymphopenia, elevated c-reactive protein, lactate dehydrogenase, D-dimer, and liver transaminases).

During the first surge of the COVID-19 pandemic in Spain in March of 2020, SARS-CoV-2 PCR testing had limited availability, and test results were delayed by 24–72 h. PCR test results of study subjects were not known at the time of study enrolment. During data analysis, subjects were categorized as having “confirmed” COVID-19 defined by a positive PCR test result or “suspected” COVID-19 defined by either a negative PCR test result or nonperformance of PCR testing.

After informing subjects about the study objectives and minimal risks, verbal consent was obtained and documented in the electronic medical record. Written consent using paper was not feasible due to the risk of fomite transmission of SARS-CoV-2 to study personnel. This study complied with the Declaration of Helsinki and was approved by the local ethics committee and hospital research committee (PI 64/20).

### 2.2. Lung Ultrasound Exam

A bedside LUS exam was performed on each subject by one of two physicians with expertise in point-of-care ultrasound (M.M.G., F.J.T.M.). Both physician sonographers performed an LUS exam on all subjects who were clinically diagnosed with COVID-19 by an attending physician in the emergency department. The LUS exam was performed independent of the evaluation by the attending physician in the emergency department. Both physician sonographers were blinded to each patient’s history, laboratory results, and radiographic images and were not directly involved in the patient’s care. PCR test results were not available until 24–72 h after presentation and were not known at the time of the LUS exam.

Two portable ultrasound machines with curvilinear transducers (Mindray M9 (Shenzhen, China) and Esaote MyLab Omega (Genoa, Italy)) were used. The ultrasound machine and transducer were covered with plastic cling film during each exam. The physician sonographers wore N-95/FFP2 masks, impermeable gowns, and two pairs of gloves. Despite the use of personal protective equipment, the physician sonographers were required to stand behind the subjects when performing the LUS exam to avoid face-to-face contact and minimize the risk of viral transmission. The chest wall skin was cleaned with an alcohol-based antiseptic solution before each LUS exam.

The LUS protocol included 5 zones per hemithorax—three posterior zones (superior, middle, and inferior) and two lateral zones along the mid-axillary line (superior and inferior) (Figure 1). A total of 10 zones were scanned per patient. Pathological LUS findings have been previously described [10,11,16]. LUS findings were categorized as normal, discrete B-lines (3 or more B-lines per rib interspace), confluent B-lines, small subpleural consolidations (<3 cm), and lobar consolidations (Figure 2). LUS findings were recorded as video clips and written descriptions were entered into a database.

### 2.3. Chest Radiographs

All CXRs were obtained by a radiology technician and interpreted by a board-certified radiologist. Two CXR views (posterior-anterior and lateral) were taken in the radiology department. The final CXR report was entered into a database for comparison with the LUS findings. A blinded third investigator with ultrasound expertise (G.G.C.) compared the LUS and CXR findings reported by the two physician sonographers and radiologists, respectively.

### 2.4. Statistical Analysis

Subjects were categorized as having suspected or confirmed COVID-19 based on the PCR testing as stated above. CXR and LUS findings were classified into three ordinal categories for each diagnostic method—disease absent (normal lung), unilateral pulmonary infiltrates, and bilateral pulmonary infiltrates. Agreement between the two diagnostic methods was calculated using the weighted Kappa statistic using the ordinal classification system. The Kappa statistic was interpreted as follows—0.20 to 0.45 moderate agreement, 0.45 to 0.75 substantial agreement, and 0.75 to 1.0 perfect agreement [19]. Sensitivity of each method was calculated, and compared using the McNemar test. Statistical analyses were performed using the frequency (FREQ) procedure in SAS (v.9.4. Cary, NC, USA: SAS Institute Inc.; 2014).

## 3. Results

One hundred and one subjects were enrolled in the study. Five subjects were excluded (three were pregnant and could not receive a CXR; two had alternative diagnoses found). Data were analyzed from a total 96 subjects with a clinical diagnosis of COVID-19.

Characteristics of the subjects are presented in Table 1. The median age of all subjects was 48 years and half were women. The most common comorbidities were hypertension, obesity, asthma, and diabetes mellitus. A majority of subjects presented with fever, cough, and dyspnea. A greater proportion of suspected COVID-19 subjects presented <7 days whereas more confirmed COVID-19 subjects presented ≥7 days. Compared to suspected COVID-19 patients, the confirmed COVID-19 subjects had a significantly lower oxygen saturation, elevated C-reactive protein, elevated lactate dehydrogenase, and lower lymphocyte count. Most confirmed COVID-19 subjects (81%) were hospitalized while most suspected COVID-19 subjects (94%) were discharged home with close monitoring.

LUS detected pulmonary infiltrates in more subjects than CXR (81% vs. 63%) with a greater difference among subjects with suspected COVID-19 (70% vs. 40%) versus confirmed COVID-19 (95% vs. 91%) (Figure 3). Among the subjects with a normal CXR but abnormal LUS exam, 20 subjects (55%) had pulmonary infiltrates detectable by LUS (Figure 4). Furthermore, most of these subjects (*n* = 12) had bilateral infiltrates that were seen on LUS but not on CXR (Appendix A). On the contrary, among the subjects with a normal LUS exam but abnormal CXR, only two had pulmonary infiltrates detected on CXR which were described as “doubtful” or “minimal” infiltrates in the medial or left basilar lung fields per the radiologist’s official report (Figure 4).

The types of LUS and CXR findings are shown in Table 1. More suspected COVID-19 subjects had a normal LUS and CXR compared to those with confirmed disease. Among all 78 subjects with LUS abnormalities, all subjects had discrete B-lines with pleural line irregularities. Half of all subjects had confluent B-lines and 43% had small subpleural consolidations (<3 cm). In confirmed COVID-19 subjects, alveolar infiltrates on CXR and discrete or confluent B-lines on LUS were more often seen compared to those with suspected COVID-19.

The distribution of pulmonary infiltrates detected by LUS versus CXR in suspected and confirmed COVID-19 subjects is shown in Figure 5 (Appendix A). LUS detected pulmonary infiltrates compared to CXR in a greater proportion of subjects in both the right (77% vs. 57%) and left lungs (67% vs. 58%). Regarding specific lung lobes, LUS detected pulmonary infiltrates more often than CXR in all lung lobes with the greatest differences in the right middle lobe (62% vs. 32%), right lower lobe (65% vs. 46%), and left upper lobe (52% vs. 35%). In all lung lobes, pulmonary infiltrates were detected more frequently in confirmed versus suspected COVID-19 subjects by either LUS or CXR.

The correlation between LUS and CXR was assessed by weighted Kappa statistic (Appendix A). A substantial level of agreement was demonstrated between LUS and CXR for normal, unilateral or bilateral pulmonary infiltrates (Κ = 0.48 (95% CI 0.34 to 0.63)), as defined by Munoz et al. [19]. Comparing normal versus abnormal LUS and CXR, the Kappa statistic similarly showed substantial agreement (Κ = 0.46 (95% CI 0.28 to 0.63)). LUS was more sensitive than CXR for detecting pulmonary infiltrates (81% vs. 63%; *p* = 0.002) using the McNemar test.

## 4. Discussion

We reported the findings of a large prospective study assessing the correlation of LUS and CXR for detection of pulmonary infiltrates in noncritically ill COVID-19 patients. A substantial level of agreement was demonstrated between LUS and CXR, and LUS detected pulmonary infiltrates more frequently compared to CXR in all subjects. Most importantly, among the subjects with a negative CXR, abnormalities were detected by LUS in more than half of these subjects.

Confirming a diagnosis of COVID-19 by laboratory testing or diagnostic imaging is challenging, especially early in the course of the disease. PCR testing is limited by availability, high false negative rate (sensitivity 65–83%), and delays in test positivity (mean 5.1 days) [2,3,4,5]. In one study, PCR test results turned positive in 21% of patients after two consecutive negative results [20]. In our study, PCR test results were not available until 24–72 h after presentation and were unknown when the LUS exam and CXR were performed. Among the diagnostic imaging modalities, chest CT scan has been reported to have the highest sensitivity (97–98%) [2,3,4,5]. However, obtaining chest CT scans on all patients with suspected COVID-19 is impracticable during a pandemic when resources are limited, and most of the world’s population lacks access to CT imaging [21]. Thus, clinicians have had to rely primarily on CXRs and LUS to detect pulmonary infiltrates to support a clinical diagnosis of COVID-19.

The LUS findings in COVID-19 have been well described in several reports [10,11,12,13,14,15,16]. However, only two small case series have reported both CXR and LUS findings in COVID-19 patients, but neither study directly compared CXR and LUS findings nor assessed the correlation of the two imaging tests [17,18]. In our study, all patients underwent both LUS and CXR upon presentation that were interpreted by blinded experts. We demonstrated a substantial level of agreement between LUS and CXR, but LUS had a higher sensitivity for detecting pulmonary infiltrates compared to CXR (81% vs. 63%). Our findings are consistent with another study reporting the sensitivity of CXR (69%) in COVID-19 patients [8]. Similar sensitivity of LUS (85%) was reported in a meta-analysis of non-COVID pneumonia studies comparing LUS to CXR or chest CT scans [22].

A key finding of our study was the ability of LUS to detect pulmonary infiltrates in more than half of the subjects with a normal CXR. Furthermore, one-third of these subjects had bilateral findings on LUS that were not seen on CXR (Appendix A). On the contrary, only two subjects had lung infiltrates reported on CXR that were not seen on LUS; however, the radiologist’s official report commented that these were “doubtful” or “minimal” infiltrates. Based on our findings, institutions with trained clinicians can develop protocols that include LUS as part of the initial bedside evaluation of suspected COVID-19 patients. Though not assessed in our study, bedside detection of pulmonary infiltrates by LUS has the potential to guide triage and treatment decisions as new therapies emerge.

In our study, disposition decisions about hospital admission versus close monitoring at home were determined using a hospital protocol independent of the LUS findings. However, a few points deserve mention from our post-hoc analysis of disposition (Appendix A). First, subjects with a normal CXR or LUS were more often discharged home. Second, though COVID-19 PCR test results were not known at the time of presentation, more confirmed versus suspected COVID-19 subjects were admitted to the hospital versus discharged home (81% vs. 15%). Most importantly, LUS detected more unilateral (25% vs. 17%) or bilateral pulmonary infiltrates (42% vs. 19%) compared to CXR in suspected COVID-19 subjects that were safely discharged home. Whether LUS is overly sensitive for detecting pulmonary infiltrates that could lead to unnecessary admission of individuals that could be safely monitored at home is an important question to address in future studies.

We recognize that our study has limitations. First, PCR testing could only be performed on approximately half of subjects in our study because laboratory testing supplies were extremely limited during the initial surge of the COVID-19 pandemic in Madrid. However, given the high false negative rates of early PCR test kits and the 24–72 h delay in obtaining PCR test results, a clinical diagnosis of COVID-19 was typically made based on close contact and supportive laboratory findings. Second, due to concerns of healthcare workers contracting COVID-19, a rapid and focused LUS exam was performed with the physician sonographer standing behind the patient and interrogating the posterior and lateral chest walls. Recent publications have recommended standardization of LUS protocols in COVID-19 to foster pooling of data from multiple institutions in future studies [14,23]. Third, chest CT scans could not be obtained in all patients with suspected COVID-19 due to limited hospital resources, and only three subjects underwent a chest CT scan.

## 5. Conclusions

In summary, LUS findings correlated well with those of CXR in patients with suspected or confirmed COVID-19. Lung ultrasound was able to detect pulmonary infiltrates in more than half of patients with a normal CXR. Thus, a LUS exam may be performed at the bedside as the initial diagnostic imaging test in patients with COVID-19. Future studies are needed to evaluate the use of a standardized LUS protocol on triage decisions and health services of patients with suspected COVID-19.

## Figures and Tables

**Figure 1 diagnostics-11-00373-f001:**
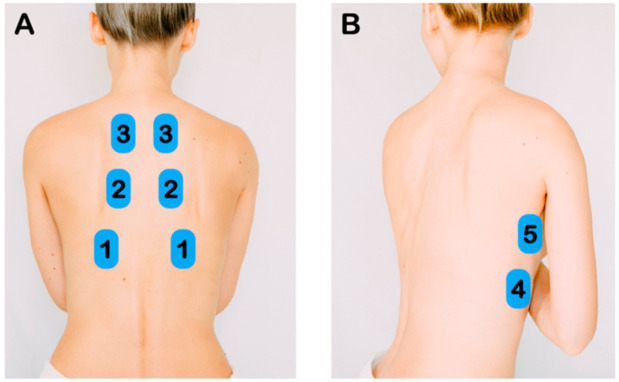
Lung Ultrasound Exam Points. (**A**) After identifying the diaphragm, the transducer was slide cephalad to image the inferior, middle, and superior zones of the posterior chest. (**B**) Along the mid-axillary line, the inferior and superior lung zones of the lateral chest were imaged.

**Figure 2 diagnostics-11-00373-f002:**
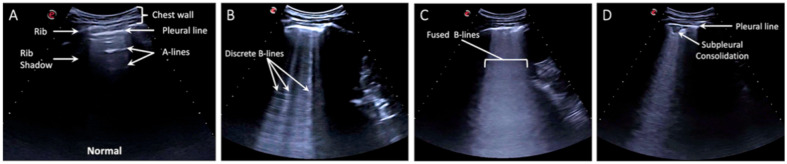
Characteristic Lung Lesions in coronavirus disease 2019 (COVID-19). (**A**) Normal lung ultrasound is defined by visualization of pleural sliding and A-lines. (**B**) Discrete B-lines are individual hyperechoic, laser-like artifacts the emanate from the pleural line and are due to increased interstitial fluid in the acute setting. Discrete B-lines are typically the first sign of COVID-19. (**C**) Fused or confluent B-lines are seen when individual B-lines coalesce as interstitial fluid increases. (**D**) Subpleural consolidations are typically small (<3 cm) areas of consolidation that are seen just below the pleural line.

**Figure 3 diagnostics-11-00373-f003:**
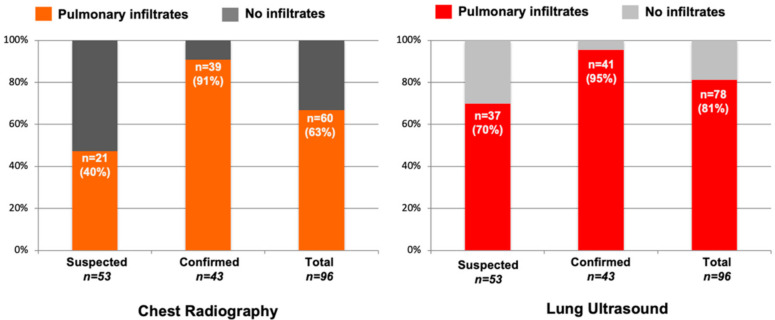
Chest X-ray and Lung Ultrasound for Detection of Pulmonary Infiltrates. The number of suspected or confirmed COVID-19 subjects (*n*) with or without pulmonary infiltrates detected by chest X-ray or lung ultrasound is demonstrated. In both suspected and confirmed COVID-19 subjects, lung ultrasound was able to detect pulmonary infiltrates more often than chest radiography.

**Figure 4 diagnostics-11-00373-f004:**
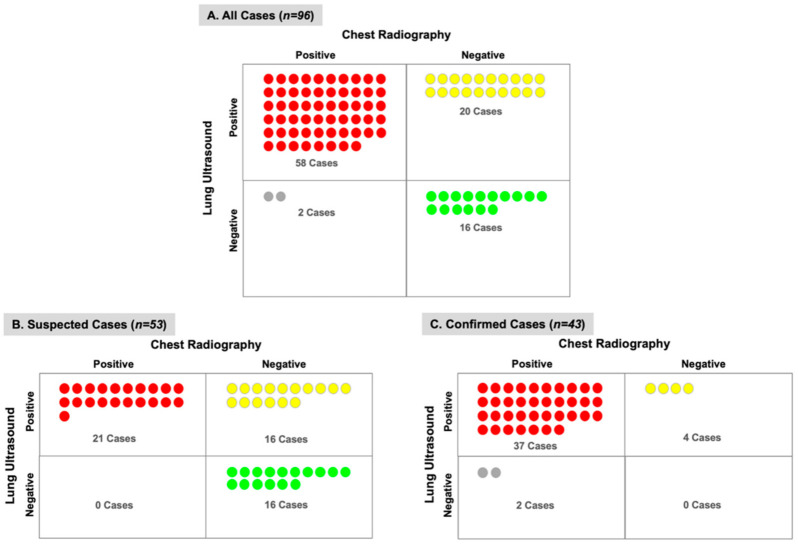
Correlation of Chest X-ray and Lung Ultrasound in Detection of Pulmonary Infiltrates. The number of subjects (*n*) and agreement between chest X-ray and lung ultrasound is shown for (**A**) all cases, (**B**) suspected COVID-19 cases, and (**C**) confirmed COVID-19 cases. Lung ultrasound detected pulmonary infiltrates in 20 subjects with a normal chest X-ray, whereas chest X-ray detected pulmonary infiltrates in 2 subjects with a normal LUS exam.

**Figure 5 diagnostics-11-00373-f005:**
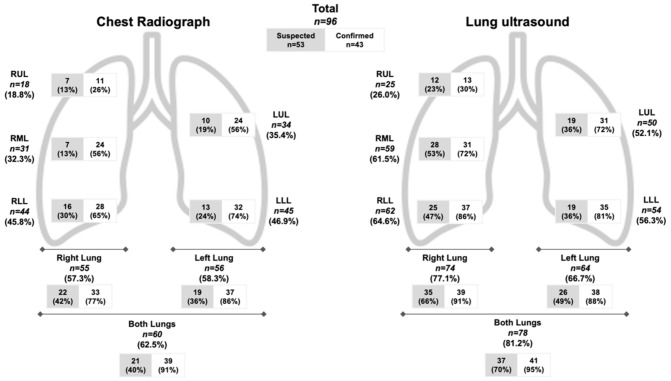
Distribution of Pulmonary Infiltrates Detected by Chest X-ray vs. Lung Ultrasound. The number of subjects (*n*) with confirmed or suspected COVID-19 who had pulmonary infiltrates detected in the upper, middle, or lower lobes of the right and left lung is demonstrated.

**Table 1 diagnostics-11-00373-t001:** Characteristics of Subjects with Suspected and Confirmed COVID-19.

Characteristic	Suspected *n* = 53 *n* (%)	Confirmed *n* = 43 *n* (%)	Total *n* = 96 *n* (%)	*p*-Value
**Gender**				0.105
Male	22 (41.5)	25 (58.1)	47 (49.0)	
Female	31 (58.5)	18 (41.9)	49 (51.0)	
**Age**				0.092
Median years (IQR)	47 (40.0–56.5)	51 (41.0–64.0)	48 (41.0–58.0)	
<30	5 (9.4)	0 (0.0)	5 (5.2)	
30–39	8 (15.1)	6 (14.0)	14 (14.6)	
40–49	18 (34.0)	14 (32.5)	32 (33.3)	
50–59	13 (24.5)	9 (20.9)	22 (22.9)	
60–69	7 (13.2)	8 (18.6)	15 (15.7)	
70–79	2 (3.8)	5 (11.7)	7 (7.3)	
≥80	0 (0.0)	1 (2.3)	1 (1.0)	
**Ethnicity**				0.433
Caucasian	28 (52.8)	29 (67.4)	57 (59.4)	
Latin American	17 (32.1)	11 (25.6)	28 (29.2)	
African	5 (9.4)	3 (7.0)	8 (8.3)	
Asian	2 (3.8)	0 (0.0)	2 (2.1)	
Other	1 (1.9)	0 (0.0)	1 (1.0)	
**Comorbidities**				
Hypertension	14 (32.6)	12 (22.6)	26 (27.1)	0.277
Obesity	11 (25.6)	9 (17.0)	20 (20.8)	0.302
Asthma	7 (16.3)	5 (9.4)	12 (12.5)	0.313
Diabetes mellitus	4 (9.3)	4 (7.5)	8 (8.3)	0.757
Coronary artery disease	1 (2.3)	2 (3.8)	3 (3.1)	0.685
Chronic obstructive pulmonary disease	1 (2.3)	2 (3.8)	3 (3.1)	0.685
Bronchitis	1 (2.3)	0 (0.0)	1 (1.0)	0.264
Human immunodeficiency virus	1 (2.3)	0 (0.0)	1 (1.0)	0.264
Other	4 (9.3)	1 (1.9)	5 (5.2)	0.104
**Symptoms**				
Fever	43 (81.1)	38 (88.4)	81 (84.4)	0.331
Cough	42 (79.2)	37 (86.0)	79 (82.3)	0.385
Dyspnea	28 (52.8)	30 (69.8)	58 (60.4)	0.092
Myalgia	19 (35.8)	11 (25.6)	30 (31.3)	0.280
Diarrhea	11 (20.8)	9 (20.9)	20 (20.8)	0.983
Headache	10 (18.9)	2 (16.7)	12 (12.5)	0.036
Sore throat	7 (13.2)	3 (7.0)	10 (10.4)	0.320
Other	3 (5.7)	1 (2.3)	4 (4.2)	0.416
**Days of Symptoms**				
Median days (IQR)	6.0 (3.0–9.5)	7.0 (5.0–10.0)	7.0 (4.0–10.0)	0.080
<7 days	31 (58.5)	13 (30.2)	44 (45.8)	0.006
≥7 days	22 (41.5)	30 (57.7)	52 (54.2)	
**Oxygen Saturation**				
Median % (IQR)	98.0 (96–99)	95.0 (94–97)	97.0 (95–98)	<0.001
**Lung Physical Examination**				0.285
Normal	25 (47.2)	25 (58.1)	50 (52.1)	
Abnormal	28 (52.8)	18 (41.9)	46 (47.9)	
**Laboratory Data**, **median** (**IQR**)				
Leukocytes (*n* = 73) (×10^3^ /µL)	6.1 (5.4–8.2)	6.8 (5.3–8.4)	6.5 (5.4–8.2)	0.696
Lymphocytes (*n* = 73) (×10^3^ /µL)	1.6 (1.2–1.9)	1.2 (0.8–1.4)	1.4 (1.0–1.7)	<0.001
LDH (*n* = 71) (Nl=120–240 U/L)	208 (160–226)	248 (208–310)	220 (184–275)	0.001
CRP (*n* = 73) (Nl < 5 mg/L)	26 (5.0–48.0)	51 (28.7–113.2)	41 (12.5–91.5)	0.001
D-dimer (*n* = 68) (Nl < 500 ng/mL)	455 (350–700)	535 (405–1052)	500 (370–757.5)	0.170
**Chest X-ray**				
Normal	32 (60.3)	4 (9.3)	36 (37.5)	<0.001
Alveolar infiltrate	15 (28.3)	36 (83.7)	51 (53.1)	<0.001
Interstitial infiltrate	9 (17.0)	12 (27.9)	21 (21.9)	0.198
Other	1 (1.9)	2 (4.7)	3 (3.1)	0.439
**Lung Ultrasound Findings**				
Normal	16 (30.2)	2 (4.7)	18 (18.8)	0.001
Discrete B-lines	37 (69.8)	41 (95.3)	78 (81.3)	0.001
Confluent B-lines	19 (35.8)	29 (67.4)	48 (50.0)	0.002
Small Subpleural Consolidations (<3 cm)	18 (34.0)	23 (53.5)	41 (42.7)	0.054
Large Consolidations (>3 cm)	2 (3.8)	0	2 (2.1)	0.198
Pleural effusion	1 (1.9)	1 (2.3)	2 (2.1)	0.881
Other	1 (1.9)	1 (2.3)	2 (2.1)	0.881
**Disposition**				<0.001
Hospitalized	3 (5.7)	35 (81.4)	38 (39.6)	
Home with Close Follow-up	50 (94.3)	8 (18.6)	58 (60.4)	

IQR, interquartile range; LDH, lactate dehydrogenase; CRP, C-reactive protein.

## Data Availability

All the data used in this study will be made publicly upon publication of the study.

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
