# Peer review of "Comparison of Lung Ultrasound versus Chest X-ray for Detection of Pulmonary Infiltrates in COVID-19"

_diagnostics, 2021, doi:10.3390/diagnostics11020373_

Round 1
Reviewer 1 Report
Authors showed very interesting approaches using wireless(point-of-care) /mobile ultrasound imaging and x-ray chest imaging in COVID-19. Authors showed well summarized data to check the true and false cases for patients. Therefore, authors can conclude that wireless/mobile ultrasound imaging could be useful for the patients in COVID-19. Authors also showed the limitation of the proposed study because COVID-19 pandemic situation makes the data be unpredictable for some cases. There are no English grammar issues at all. However, some Figure qualities need to be improved. Therefore, the manuscript can be minor revision. There are some comments as suggested.
- Please check MDPI reference styles.
- Figure A1 labels are too small so authors need to increase the size of Figure A1.
- Please do not use reference numbers before . in the sentences.
- Please correct "We report" to "We reported".
- Please correct "has limitations." to "has some limitations".
Author Response
Please see attached document with responses to reviewer comments.

Reviewer 2 Report
I would like to congratulate the authors for the study carried out and for the amazing work.
My comments to the authors are:
There are several misspelled words in the manuscript. For example minimise (line 103); categorised (line 108); dyspnoea (line 143); recognise (line 234). Check all paper
Section 2.5 is not required
Figures must be included in the text
It´s need a conclusion section.
References
The style of the references is not correct.
Author Response

(The authors gave the same response as above.)
